# Ragging, a Form of University Violence in Sri Lanka—Prevalence, Self-Perceived Health Consequences, Help-Seeking Behavior and Associated Factors

**DOI:** 10.3390/ijerph19148383

**Published:** 2022-07-08

**Authors:** Ayanthi Wickramasinghe, Birgitta Essén, Shirin Ziaei, Rajendra Surenthirakumaran, Pia Axemo

**Affiliations:** 1Department of Women and Children’s Health, International Maternal and Child Health Unit (IMCH), Uppsala University, 75185 Uppsala, Sweden; birgitta.essen@kbh.uu.se (B.E.); shirin.ziaei@kbh.uu.se (S.Z.); pia.axemo@kbh.uu.se (P.A.); 2Department of Community and Family Medicine, Faculty of Medicine, University of Jaffna, Jaffna 70140, Sri Lanka; surenthirakumaran@gmail.com

**Keywords:** hazing, harassment, abuse, violence, university students, public health

## Abstract

Ragging is an initiation ritual practiced in Sri Lankan universities for generations, although research is scarce. This practice has several adverse consequences such as physical, psychological, and behavioral effects and increased university dropouts. The aim of this study was to investigate the prevalence of different types of ragging: emotional/verbal, physical and sexual ragging, self-perceived health consequences, help-seeking behavior, and factors associated with the experience of ragging. A cross-sectional study was conducted among 623, second- and third-year students from the medical, and technology faculties in Jaffna University. Bivariate associations were assessed using chi-squared tests. Logistic regression was used to evaluate factors associated with any type of ragging. Ragging was experienced by 59% of the students, emotional/verbal ragging being the most common. A total of 54% of students suffered one or more health consequences and mainly sought help from friends and family, with few seeking formal help. Factors associated with any type of ragging were faculty and year of study. This study emphasizes the urgent need to address this public health problem. It is important that there are adequate student support services, planning and implementation of effective interventions, as well as ensuring that existing policies are strengthened, to reduce or eliminate ragging in Sri Lanka.

## 1. Introduction

Universities should be safe havens where students can grow, learn, and thrive in a secure environment. Often, students’ wellbeing is affected due to initiation practices carried out in educational institutions over the world, and known as “Hazing” in North America, “Bizutage” in France, “Praxe” in Portugal, “Mopokaste” in Finland, and “Ragging” in South Asian countries [1,2]. These practices can range from being welcoming and consensual to violent and dangerous, depending on the context [3]. They are part of a complex social phenomenon and are contextual depending on the country, people, and situation [4]. In the Western world, these practices, especially within fraternities and sororities, are often of a sexual nature which could lead to sexual abuse. They can also include drinking games and binge drinking to form bonds [5,6]. Some of these rituals consist of harassment, intimidation, and humiliation [7]. Undergoing these rites is seen as essential to be accepted by seniors and their peer group [8]. This practice is similar to the initiation into other groups such as the military, sport teams, and gangs [5]. International studies conducted on hazing, in clubs and college organizations in universities demonstrate the prevalence to range from 37–78% [5,7].

The initiation practice carried out in South Asian countries such as Sri Lanka, India, Nepal, and Pakistan are referred to as “ragging” [9,10,11,12,13]. It often occurs during the first few months after entering the University, the so-called “ragging period” [14,15], whereby students are meant to get acquainted and to increase bonds between the new entrants and the senior students. This tradition has now escalated to a more menacing form where the junior students are subjected to psychological, physical, or sexual violence, often resulting in severe health consequences [11,16,17]. In Sri Lanka, ragging is defined as: “any deliberate act by an individual student or group of students, which causes physical or psychological stress or trauma” [18].

Students who experienced ragging are often unwilling to speak about this practice or make complaints due to the fear of being ostracized [19,20]. New students who make complaints are isolated from their classmates, stigmatized, and branded as “*anti-raggers*”. They are excluded from all university functions and parties organized by the students, not given leadership roles, and the rest of their peers do not talk or socialize with them [15,20].

Sri Lanka is a multiethnic, multicultural, and multilingual country consisting of an ethno-religious blend of Sinhalese (75%), Tamils (11%), Moors (Muslims) (9%), and other groups (5%) [21] and is regionally segregated. Tamil Hindus predominate in the northern province, and Sinhalese Buddhists predominate in the rest of the country, while the Moors are not limited to a certain area. The official languages are Sinhala and Tamil with English as the bridging language.

Sri Lanka upholds a hierarchical social structure with patriarchal values [15]. In the past, violence has been a way to foster discipline as evidenced by the continued presence of corporal punishment at home despite it being officially forbidden in schools in 2005 [22]. Violence, seen as a symbol of power in this male-centric society, is apparent in the high prevalence rates (18–72%) of different types of intimate partner violence (IPV) [23]. A study conducted in four districts in Sri Lanka found that 36% of ever-partnered men had perpetrated physical or sexual IPV [24].

Since 1998, ragging has been a criminal offense in Sri Lanka and carries severe punishment from 2 to 10 years imprisonment [25]. Due to the harsh the punishment, this law has rarely been enforced. Nevertheless, the university grants commission (UGC), the university administrative body, has issued several guidelines and circulars to facilitate the enforcement of the law. The UGC has also created several methods to lodge a complaint, such as a hotline, an Internet portal, and a mobile application.

Ragging has been found to cause several adverse health effects including depression and other psychological and behavioral consequences [11,16,26]. There have been several cases in Sri Lanka where students were severely injured, paralyzed, or killed as a result of ragging [14]. According to reports by the Ministry of Education, approximately 2000 students dropout annually, and several students have committed suicide as a consequence of ragging [27]. Although the government considers ragging a significant public health problem in Sri Lanka, research conducted on ragging is limited. Baseline data and prevalence studies on ragging are needed to guide efforts to end ragging by providing knowledge for planning and implementing effective interventions.

The hypothesis of the study was that the age of the faculty was associated with the prevalence of ragging: therefore, the prevalence of ragging would be higher in the faculty of medicine as it is a well-established faculty with preexisting student norms compared to the faculty of technology which is a newly formed faculty with the third-year students being the first batch of students.

In this study, we aim to investigate the prevalence of different types of ragging, including emotional/verbal, physical, and sexual ragging. The study examines self-perceived health consequences, help-seeking behavior, and factors associated with the experience of ragging among second and third-year university students in the faculties of medicine and technology in the University of Jaffna.

## 2. Materials and Methods

### 2.1. Study Setting

The University of Jaffna is situated on the northern peninsula. This was a war zone during the armed conflict from 1983 to 2009 between the Sri Lankan government and the Liberation Tigers of Tamil Eelam (LTTE). The university was established in 1974 [28] and has almost similar numbers of students of each ethnic background unlike other universities where the majority are Sinhalese. Hence, the University of Jaffna was chosen as the study site due to the student composition.

The study was conducted among the students in medical and technology faculties. The medical faculty was chosen as it is well-established, with longstanding traditions and cultural norms. The medical faculty, established in 1978, is housed adjacent to the main campus in the town of Jaffna. In contrast, the technology faculty was established in 2016 and is situated in an isolated area 80 km from the main campus in the Killinochi district, with evolving student norms.

### 2.2. Study Population

Ragging is believed to be confined within faculties. The victims of ragging are generally the first-year students, and the perpetrators are senior students from the second year onwards. The first-year students were excluded to prevent re-traumatization as they could still be affected by ragging experiences. All the students from the second and third year of the medical and technology faculties were invited to participate in the study. The second-year students were chosen as they could have been both victims and perpetrators of ragging. The third-year technology faculty students were chosen as they were the first batch of students that had no seniors preceding them.

### 2.3. Study Design

The study was a cross-sectional survey using a self-administered questionnaire with 41 questions evaluating experiences of ragging (emotional/verbal, physical, and sexual), as well as self-perceived health consequences and help-seeking behavior.

This study was the point of initiation for a larger study conducted among the students of Jaffna University. This baseline study was one of the two quantitative studies conducted, and the other was on the prevalence of depression among students. These studies were supplemented by two qualitative studies exploring student perceptions on ragging [20] and the perceptions of lecturers and other staff attached to Jaffna University.

The questionnaire was previously validated in Sri Lanka in a study among dental students [11] and was adapted from a survey instrument measuring abuse amongst Canadian medical students [29]. The tool was found to comprise relevant questions with good reliability and internal consistency with Cronbach’s alpha = 0.86 and 0.86 for the experience of ragging and self-perceived health consequences, respectively, in this study.

The questionnaire was distributed to the medical students in English as this was their language of education. The questionnaire was translated into Sinhala and Tamil by the research group, as the level of English comprehension was lower in the faculty of technology. The data collection was carried out in February 2019 in the medical faculty by the principal investigator, but due to the closure of the technology faculty caused by a serious ragging incident, data collection in this faculty was carried out in September 2019 by a research assistant. After explaining the aim of the study and obtaining consent, the questionnaires were distributed at the end of a compulsory lecture. Students were informed that participation was completely voluntary, and they could opt out of any questions they were uncomfortable answering. The principal investigator was present to make clarifications in English and Sinhalese, and a research assistant answered questions in Tamil. Questionnaires contained a serial number to protect the identity of the participants and ensure anonymity.

#### Variables

The questionnaire had sections on demographic data, experience of ragging, details of the ragging incident such as perpetrator and place, self-perceived health consequences, and help-seeking behavior.

There were several questions on students’ background characteristics. Sex was categorized as male or female. Ethnicity was classified as Sinhalese, Tamil, and Moors/Muslim. Father’s/mother’s education was categorized to below 11 years of schooling (compulsory years of schooling) and above 11 years of schooling. Father’s occupation was categorized into self-employed, full-time employed, and unemployed/irregular employment and mother’s occupation into employed (self-employed or full-time employed) and home maker/irregular employment.

Students were asked if they had experienced any type of ragging, such as emotional/verbal, physical, or sexual ragging during their time at the university. Questions on emotional/verbal ragging included: have you ever been shouted at, been given duties or work as punishment, been ignored, treated differently, threatened, or any other type of emotional/verbal ragging. Questions on physical ragging included: have you ever been slapped, pushed, made to eat/drink something unpleasant, punched, kicked, or suffered any other type of physical ragging. Questions on sexual ragging comprised: have you ever been stared at, received unwanted sexual comments, made fun of sexual habits, shown pornographic material, had unwanted sexual advances, asked to engage in sexual acts for rewards, had your clothes removed by force, experienced unwanted sexual touching, forced to have intercourse, or any other type of sexual ragging. Students were considered to have been exposed to any type of ragging if they responded positively to any question on ragging.

Students were asked to respond to the questions on self-perceived health consequences if they had experienced any form of ragging while at the university. These included questions such as did you ever experience: upsetting memories, upsetting dreams, reduced interest for studies, insomnia, avoiding situations or activities, being irritable/outbursts of anger, feeling lonely, missing classes, and/or avoiding social activities as a result of ragging. Participants were believed to have suffered health consequences if they responded positively to one or more questions. Help-seeking behavior was identified by asking the students who they reported these incidents to if they had experienced any form of ragging. The options were, family and friends, lecturers, student counselors, the deans, and police and/or the UGC help lines. If students responded positively to one or several statements, they were considered to have sought help.

### 2.4. Analysis

Descriptive characteristics of the students as well as experience of different types of ragging, self-perceived health consequences, and help-seeking behavior are presented as frequencies and percentages. Chi-squared tests were used to compare proportions and to evaluate bivariate association between characteristics of the students and their experience of any type of ragging. Logistic regression models were used to evaluate factors associated with experience of any type of ragging among the students. The results are presented in two models. The first unadjusted model was followed by a second model including all factors considered significant in univariate analysis with *p* < 0.20. Odds ratios (OR) with 95% confidence intervals were calculated. Statistical significance was considered if *p* < 0.05. The data were analyzed using the statistical software, “RStudio” (3.5.2 Eggshell Igloo, RStudio, Boston, MA, USA).

## 3. Results

From the total number of students (*n* = 683) in both faculties, 623 students completed the questionnaires, giving a response rate of 91%. Among the medical students 118 (*n* = 149) of the second-year students, and 128 (*n* = 138) of the third-year students responded. While 190 (*n* = 200) students in the second year, and 187 (*n* = 196) students in the third year responded in the technology faculty. The percentage of female students (53%) was higher than the male students. The age span was between 21and 28 years, with students ≥24 years being more likely to experience any type of ragging (*p* = 0.001). More than half of the students were Sinhalese. Most of the students have parents with more than 11 years of education. Among the students, 43% had fathers with full-time employment, while most mothers were homemakers (68%). More than 60% of the students belonged to the technology faculty. Students from the medical faculty (*p* < 0.001) and second-year students (*p* = 0.002) were more likely to experience any type of ragging (Table 1).

### 3.1. Type of Ragging

Among the students, 59% experienced at least one type of ragging (Table 2). Emotional ragging was the most experienced form of ragging, with 40% of students experiencing at least one form of emotional/verbal ragging. When stratified by sex, physical ragging was seen among 75% of male students (*p* > 0.05). No other significant differences were found between the sexes with regards to other types of ragging (data not shown). Stratification by ethnicity did not yield any significant results.

### 3.2. Self-Perceived Health Consequences and Help-Seeking Behavior

Among the students who experienced any type of ragging, 54% reported at least one type of self-perceived health consequences (Table 3). Irritability/outbursts of anger were the most common reported experience (33%), followed by upsetting memories (27%) and avoiding situations or activities (27%).

A large proportion of students (57%) had sought some form of help, commonly from friends and family (Table 3). The formal channels of help such as university staff, except for student counsellors, police, and UGC helplines were underutilized.

### 3.3. Factors Associated with Ragging

In the unadjusted analysis, maternal level of education, faculty, and year of education were significantly associated with experience of ragging among the students (Table 4). Students who had mothers with more than 11 years of education had higher odds of experiencing any type of ragging compared to the students who had mothers with less than 11 years of education (UOR 1.52, 95% of CI: 1.04–2.21). Students belonging to the technology faculty (UOR 0.39, 95% of CI: 0.27–0.55) and students in the third year (UOR 0.60, 95% of CI: 0.43–0.82) had significantly lower odds of experiencing any type of ragging compared to the students in the medical faculty and second-year students respectively. In the adjusted model, maternal education was no longer associated with experience of ragging (AOR 1.36, 95% of CI: 0.83–2.24). The lower odds of experiencing ragging among the students of the technology faculty (AOR 0.44, 95% of CI: 0.29–0.63) as compared to the medical faculty remained significant in the adjusted model. Similarly, the decreased odds of being ragged in the third year (AOR 0.67, 95% of CI: 0.46, 0.96) compared to the second year was maintained significant after adjusting for other factors.

## 4. Discussion

Our findings demonstrate that as many as 59% of the students undergo one or several forms of ragging in the medical and technology faculties. Students from the recently established faculty of technology were less exposed to ragging than the students from the well-established medical faculty which may be due to the time required to develop student norms. Another reason could be that ragging is usually carried out by the senior students of each faculty [20], and since the third-year students were the first batch in the technology faculty, they had no seniors above them to carry out ragging in the same faculty. The medical faculty, being in existence for over 40 years, has established practices and student cultural norms that continue each year. The medical faculty is also situated in the main campus along with most of the other faculties in the university of Jaffna, where ragging is known to occur. This may have contributed to the lower prevalence of ragging among the technology students compared to the medical students. However, there were some students from the technology faculty who were ragged by senior students from the two other faculties situated in that location. This may have occurred due to the isolated location of the technology faculty, giving rise to an opportunity for students from another faculty to rag these students.

A spectrum of ragging exists in Sri Lankan universities, from mild forms where new students must follow a certain dress code to more severe forms such as physical, sexual, and emotional/verbal [15]. New students should undergo ragging as a part of their initiation, and if they refuse to accept this so-called “sub-culture”, they are socially isolated throughout their time at the university. Among the medical students, 72% reported experiencing ragging. Although half of the students at the newly established technology faculty underwent ragging, it was less than the medical students. Similarly, a prevalence study among 65 Sri Lankan dental students found that 50% of the students had experienced some form of ragging [11]. These finding are in accordance with the well-known but understudied phenomenon of ragging that appears prevalent in Sri Lankan universities [14]. Quantitative studies on university violence carried out in Nepal, Pakistan, and Portugal showed a prevalence of 52–78% [7,9,12].

Our study demonstrates a significant proportion of the students undergo several forms of ragging. These findings are similar to the findings of other studies on harassment, bullying, and hazing [5,9,11,30]. Emotional/verbal ragging was the most common form of ragging, which another research from South East Asia also demonstrated [11,12,13,31]. Senior students in our study often intimidated and demeaned their juniors by shouting, punishing, and threatening them as a part of their initiation.

Physical ragging was reported to a lesser extent than emotional/verbal ragging in our study and several others [11,17]. Physical violence occurring among university students in the USA was found to cause injuries and death [6,32], and another study in Colombia [33] demonstrated the socio-economic and socio-political impact led to increased physical violence in bullying. Sri Lanka is recovering from a 30-year civil war, and students born during this era have grown up with violence being a part of their lives and have most likely been directly subjected to violence such as corporal punishment in schools and at home. They have also frequently been exposed to violence occurring through media and movies, resulting in the normalization of violent behavior which could have led to underreporting. Studies from other countries confirm this [34,35,36].

Participants admitted to experiencing milder forms of sexual ragging such as unwanted sexual comments. In Sri Lanka, sex is a taboo subject not discussed in public. Sex education is not offered in schools, premarital sex is unacceptable. and chastity is held in high regard. These factors can contribute to students’ unwillingness to answer questions related to sexual ragging [15]. Therefore, underreporting incidents of sexual violence due to embarrassment and social stigma could occur. Serious incidents of sexual abuse as a part of ragging have been reported in local newspapers [37,38,39,40,41,42]. Studies carried out in Sri Lanka [11], Nepal [12] and in the USA [32] demonstrated sexual violence to be a part of initiation practices among university students. Research in universities in Norway [40], Canada [41], and South Africa [42] revealed a higher prevalence of sexual harassment than our study.

The questionnaire did not cover all the health outcomes which could occur due to ragging as we only asked for self-perceived health problems. However, our results revealed that more than half of the students suffered one or more non-somatic health consequences. It has been demonstrated that if students suffer ragging in universities, the negative effects could give rise to self-perceived ill health such as anxiety, depression, and several other psychological and behavioral disorders [11,26,43,44]. This may also cause students to drop out of universities. Suicides and deaths linked to ragging have been reported in the Sri Lankan media. A study on global newspaper reports of deaths linked to hazing, bullying, and ragging between 1950 and 2007 indicated around 250 deaths among students were linked to these practices, and the reported deaths had increased in the last 10 years [45].

Our study findings indicated that 57% of students sought formal or informal help. Students frequently discussed the incident with friends and family. This was on par with help-seeking behavior in other studies [5,46,47]. Although several ways to report incidents of ragging and seeking help exist, underreporting could occur due to the lack of trust in the system and knowing that the perpetrators of ragging are rarely punished. Several researchers indicated [30,48] that the lack of self-identification of harassment by the new students may hinder their help-seeking behavior. Several studies reported that students believe ragging increases comradery [5,30] between each other, while some claim to enjoy this practice [12,49]. Students not identifying ragging as a harmful practice and claiming they enjoy it may be due to ragging being accepted as a part of the university sub-culture that students need to experience as a part of university life. Being ostracized from their peers and appearing to be weak may also be a reason this practice has endured and it rarely discussed, which in turn leads to underreporting and lower help-seeking behavior.

Gender differences are known to play a role in ragging [4,11,32,50]. Harassment of students is considered a gender phenomenon occurring in a social context fueled by gender disparities [7]. In our findings, gender was found to be insignificant although studies from the USA and Pakistan found males students to be more harassed [4,9,48,51]. The inconsequentiality of gender in our study could be due to the very gender roles adopted by Sri Lankan males, whereby they often do not share experiences that portray them as weak [11,15].

Sociodemographic factors did not play a role in our study. Father and mother’s level of education and occupation were used as a proxy to explore the socioeconomic status of the students. The univariate analysis indicated students with more educated mothers were at increased odds of experiencing ragging, although this significance was lost in the multivariate analysis. This may indicate that students with better socioeconomic status were more victimized. In Sri Lanka, students are known to use ragging as a way to equalize the social and economic hierarchies [20,52]. Similarly, a multi-country study [53] and a study from Colombia [33] demonstrated the larger economic inequalities among students in educational institutions increased the risk of violence. Furthermore, peer violence was more accepted among youth from lower socioeconomic strata [53]. Ethnic background was not associated with the risk of ragging. Underreporting to maintain the secrecy of this ritual and providing politically correct answers may mask an increased prevalence among some groups.

Students’ experience of ragging was linked to the faculty. The students belonging to the newly formed technology faculty situated a large distance away from the main campus were at less risk of experiencing ragging compared to the well-established medical faculty. Although it is believed that student norms take time to develop and evolve, especially in practices such as ragging, it is understudied, and literature is scare. The year of study appeared to be relevant, with third-year students encountering less ragging. These results are comparable to the existing information on ragging, which describes that, junior students experience more ragging which is perpetrated by senior students [14,15].

### Limitation and Strengths

As this is a cross-sectional study, we cannot determine causal relationships but can only discuss associations. Since this study topic is sensitive and victims may become future perpetrators as senior students, they may refrain from revealing information about ragging, leading to underreporting. As ragging may have victimized first-year students in this study, recall bias may be a possibility. The questionnaire was distributed to the technology faculty some months after the campus had been closed due to a severe ragging incident. This could have led to underreporting. Students were not asked about abuse before entering the university which would have helped establish a history of violence. Strengths were that this study was conducted in two faculties, one old and one newly established, with students from two different years. There was a high response rate. The university of Jaffna was chosen because it had the most equal distribution of students of each ethnicity. Although our results cannot be generalized, these findings may apply to other universities around the country. Though using a previously validated questionnaire was a strength, it had certain limitations. For example, the questions on self-perceived health consequences only pertained to the psychological aspects and did not include any somatic symptoms, self-induced harm, or suicidal thoughts that may occur due to ragging.

## 5. Conclusions

After entering Jaffna University, 59% of the students experienced ragging. Students most frequently experienced emotional/verbal ragging, and more than half reported suffering from one or more self-perceived health consequences and sought help. The findings imply that there is a difference in the prevalence of ragging according to the duration the faculty has been in existence. Medical students belonging to an old well-established faculty seemed to be subjected to more ragging compared to the technology students. The second-year students were more likely to experiencing ragging. The high occurrence of ragging among the students indicates the seriousness of this public health problem and the need to address this social issue. Providing adequate student support services, planning interventions, and implementing existing policies are required to reduce ragging and put an end to university violence.

Interventions should be aimed at creating a better understanding of the harmful consequences of ragging. New students should be targeted to become agents of change by showing them that the ragging “subculture” causes more harm than good and should not be a part of university life. It is essential to let students know that they have the support of the university administration and staff in this process. Then students can be encouraged to break the silence surrounding this practice and bring about the necessary cultural change to curb this detrimental practice.

There is also a need to conduct further studies on the mental health consequences in other faculties, as well as other universities. Additional explorative qualitative studies among students and staff are necessary to complement and further understand the many underlying mechanisms of the phenomenon of ragging.

## Figures and Tables

**Table 1 ijerph-19-08383-t001:** Descriptive characteristics of the second- and third-year students from the faculties of medicine and technology.

Variable	Number of Students (*n* = 623)	Experienced Any Type of Ragging (*n* = 366 *)	Did Not Experience Ragging (*n* = 257 *)	*p*-Value **
**Sex**				
Female	331 (53%)	191 (58%)	140 (42%)	0.57
Male	292 (47%)	175 (60%)	117 (40%)	
**Age group**				
21 years	82 (13%)	51 (62%)	31 (38%)	**0.001**
22 years	266 (43%)	165 (62%)	101 (38%)	
23 years	222 (37%)	111 (50%)	111 (50%)	
≥24 years	45 (7%)	35 (78%)	10 (22%)	
**Ethnicity**				
Moors/Muslim	80 (13%)	49 (61%)	31 (39%)	0.25
Sinhalese	322 (52%)	179 (56%)	143 (44%)	
Tamil	221 (35%)	138 (62%)	83 (38%)	
**Father’s education**				
<11 years	158 (26%)	85 (54%)	73 (46%)	0.10
>11 years	453 (74%)	277 (61%)	176 (39%)	
**Father’s occupation**				
Self-employed	161 (27%)	101 (63%)	60 (37%)	0.45
Full-time employed	263 (43%)	149 (57%)	114 (43%)	
Unemployed /Irregular employment	185 (30%)	107 (58%)	78 (42%)	
**Mother’s education**				
<11 years	146 (23%)	75 (51%)	71 (49%)	**0.03**
>11 years	462 (75%)	285 (62%)	177 (38%)	
**Mother’s occupation**				
Employed	196 (32%)	122 (62%)	74 (38%)	0.28
Homemaker/Irregular employment	416 (68%)	240 (58%)	176 (42%)	
**Faculty**				
Medicine	246 (39%)	177 (72%)	69 (28%)	**<0.001**
Technology	377 (61%)	189 (50%)	188 (50%)	
**Year**				
Second year	308 (49%)	200 (65%)	108 (35%)	**0.002**
Third year	315 (51%)	166 (53%)	149 (47%)	

* Total sample size varies due to the missing values. ** Chi-squared was used to compare proportions. Values that are significant at the *p* < 0.05 level are shown in bold.

**Table 2 ijerph-19-08383-t002:** Types of Ragging experienced by the second- and third-year students from the faculties of medicine and technology.

Variable	Number (Percentage of Total Sample) (*n* = 623)
**Emotional/Verbal Ragging**	
Yes	252 (40%)
No	371 (60%)
**Physical ragging**	
Yes	71 (11%)
No	552 (89%)
**Sexual ragging**	
Yes	80 (13%)
No	543 (87%)
**Any type of ragging**	
Yes	366 (59%)
No	257 (41%)

Students could respond positively to more than one statement.

**Table 3 ijerph-19-08383-t003:** Self-perceived health consequences and help-seeking behavior reported by students who had experienced any of type ragging in the second and third year in the faculties of medicine and technology.

Variable	Percentage within Students Who Were Ragged (*n* = 366 **)
**Any self-perceived health consequences**	197 (54%)
Upsetting memories	100 (27%)
Upsetting dreams	48 (13%)
Reduced interest for studies	93 (25%)
Insomnia	93 (25%)
Avoid situations or activities	98 (27%)
Irritable/outbursts of anger	120 (33%)
Feeling lonely	95 (26%)
Missing classes	69 (19%)
Avoiding social activities	70 (20%)
**Any type of help seeking**	210 (57%)
Family & Friends	189 (52%)
Lecturers	40 (11%)
Student counselors	34 (10%)
Dean	22 (6%)
Police	15 (4%)
UGC * portal	8 (2%)
UGC * phone	10 (3%)
UGC * ragging mobile application	4 (1%)

* UGC—University grants commission; ** Total sample size varies due to the missing values. Students could respond positively to more than one statement.

**Table 4 ijerph-19-08383-t004:** Factors associated with experience of any type of ragging among the second- and third-year students from the faculties of medicine and technology.

Variable	UOR	95% CI	AOR	95% CI
**Age**				
21 years	Reference	Reference	Reference	Reference
22 years	0.99	0.59, 1.65	1.04	0.60, 1.76
23 years	0.61	0.36, 1.02	0.68	0.39, 1.17
>24 years	2.13	0.95, 5.08	2.17	0.89, 5.72
**Father’s education**				
<11 years	Reference	Reference	Reference	Reference
>11 years	1.35	0.93, 1.94	0.97	0.59, 1.56
**Mother’s education**				
<11 years	Reference	Reference	Reference	Reference
>11 years	**1.52**	**1.04, 2.21**	1.36	0.83, 2.24
**Faculty**				
Medical	Reference	Reference	Reference	Reference
Technology	**0.39**	**0.27, 0.55**	**0.44**	**0.29, 0.63**
**Year**				
Second year	Reference	Reference	Reference	Reference
Third year	**0.60**	**0.43, 0.82**	**0.67**	**0.46, 0.96**

UOR-unadjusted odds rations, AOR-adjusted odds ratio; CI-confidence interval; AOR (CI) values that are significant at the *p* < 0.05 level are shown in bold.

## Data Availability

The data presented in this study are available on request from the corresponding author. The data are not publicly available due to sensitive nature of the study. Although students may not be identifiable individually, groups of students maybe identifiable thereby exposing them to adverse repercussions from senior students.

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
