# Peer review of "Ragging, a Form of University Violence in Sri Lanka—Prevalence, Self-Perceived Health Consequences, Help-Seeking Behavior and Associated Factors"

_ijerph, 2022, doi:10.3390/ijerph19148383_

Round 1
Reviewer 1 Report
The article represents an interesting contribution to the topic of Ragging, a form of university violence in Sri Lanka; prevalence, self-perceived health consequences, help seeking behavior and associated factors.
It provides quantitative data that could be taken as a reference on the subject, as well as the results presented. Statistical methods, in addition to being valid, have been applied in correspondence with data processing. The three instruments used refer to the methodological relevance and contribute to validating their application in other investigations. The literature conveniently supports the study. The discussion reveals important results on the subject and reinforcement with previous studies, as well as the conclusions can be important contributions to public health, highlighting the importance of the existence of adequate support services for students, especially university students.
Finally, it is positively valued that the authors have indicated the limitations of the study, to improve their future research. And in turn, they highlight the positive aspects of their research. other studies.
For all these reasons, I consider that the article should be published.
Reviewer 2 Report
Ragging as a form of violence is an unpleasant experience that must be highlighted. This issue is treated adequately by the authors in a comprehensive way.
Regarding some needs on the text, I suggest:
Titles of the tables should be shorter and avoid repetitions such as “university of Jaffna” (lines 187, 198, 209, 229).
The discussion about the results in the Medical and Technology faculties is poor. Significative differences between students deserve a deeper thought than “due to the time required to develop student norms”. Attitudes towards power and hierarchy? Superiority feeling? Future research must be done anyway (line 237).
Match font size in the text (line 309).
Follow references rules and punctuation (avoid : after authors). Line 380 and following.
Reviewer 3 Report
The topic dealt with in this paper is very important for young people, not only in the country or in the context in which the research is carried out, but also more generally. For this reason it is considered important to disseminate this type of studies as a scientific basis for addressing a relevant problem for the formation of the civic sense of future citizens.
The biggest obstacle, however, is a cultural one. Specifically, if the topic does not emerge because it is considered taboo, it is difficult to carry out research without bias.
Regarding the research work presented, the introductory part refers to the sensitivity of the population in general and to the characteristics that may justify some results obtained (or not obtained) but it should be better connected through the part of the discussion in order to have an overall more coherent paper. .
From a methodological point of view, the authors are advised to investigate precisely the non-responses and the characteristics of the individuals who did not respond to part of the questionnaire because perhaps, in this particular case, the reasons for the non-response are more important than those of the response.
Finally, it is precisely the non-responses that can give input to understand how to sensitize young people not to act as actors and to report as victims. Only a radical cultural change can help in this matter and can only start with the youngest. Therefore, it is advisable to include in the paragraph of the conclusions possible awareness-raising actions to favor this cultural change. This could be a very important social fallout of this research.
T
Reviewer 4 Report
The reviewer has examined the submitted paper titled “Ragging, a form of university violence in Sri Lanka; prevalence, self-perceived health consequences, help-seeking behavior, and associated factors” with keen interest. The study focuses on the key problem of school violence in Sri Lanka. Violence in universities can not only seriously affect the mental health of young students but also lead to losses for society. However, the reviewer feels that the paper has not substantiated the argument sufficiently and has not yet achieved the quality required for publication. Therefore, they do not recommend the submitted paper for publication.
The submitted paper resembles a research report rather than an academic paper because it is overly descriptive. Although the study has clarified several interesting findings concerning school violence and mental health problems at Sri Lankan universities, they have not been examined theoretically or analytically by the authors. For instance, following the Introduction, the authors begin an explanation of the materials and methods of their study without presenting any theoretical background or specific hypotheses concerning the associations between ragging and mental health problems. To resolve social problems, researchers must specify the mechanisms underlying such problems and recommend specific prescriptions to address them. However, the authors’ efforts to do so in the paper are not evident.
The authors have demonstrated differences in the frequency of ragging between the two universities and attributed this variation to the differences in the duration of each university’s history. However, according to the reviewer, the two universities also offer two distinct specialized majors (medicine and technology, respectively) and have different locations (city or locality). In other words, the differences in the frequency of ragging between the two universities may have arisen not because of the different durations of their history but the different majors (or locations). As the authors did not control for the possible effects of other factors related to the frequency of ragging, their explanations referring to the different durations of university history may not be justified.
The study finds a statistically significant effect of highly educated mothers on ragging experiences. However, the authors have not discussed why having a highly educated mother has a significant and positive effect on ragging experiences. Thus, the significance of this finding is not expressed in the submitted paper. The reviewer recommends that the authors examine this interesting finding both theoretically and empirically.
In the Discussion section, the authors have argued for gender differences in ragging or differences based on other sociodemographic factors by referring to previous studies. However, these arguments have not been justified by the analytical results of their study and fail to be persuasive. Certainly, gender differences (or differences brought about by other sociodemographic factors) in ragging are salient issues in the literature on school violence. However, the reviewer believes that the authors should not discuss them in the paper without specific survey-based evidence.
Finally, if the authors successfully specify how and why the experiences of ragging and mental health problems are mutually associated, the submitted paper may contribute significantly to the literature. To do so, the reviewer believes that the authors should examine the differences in mental health between students who have undergone ragging and students with no such experience.
As the study offers some significant implications for the body of research, the reviewer hopes that the authors will theoretically and analytically improve their study in line with the comments. Thank you very much for the opportunity to review this manuscript.
Round 2
Reviewer 3 Report
The corrections made to the paper have slightly improved the setting. The reasons for weakness already outlined still remain, however the issue is important and requires visibility that can contribute to raising awareness and educating the youth population. For these reasons I think it is important to publish this and also other works on these issues.
Reviewer 4 Report
Thank you for giving an opportunity to review the revised paper. The reviewer carefully examined the paper and confirmed that the paper addressed the reviewer’s concerns, but not sufficiently. Especially, he was concerned about whether the paper achieved the quality required for publication on high quality journal. For instance, the authors themselves recognize that the paper is situated as just only a baseline study on the prevalence of different types of ragging.
On the other hand, the reviewer thinks that the paper has some considerable merits on studies in school violence and might contribute to the literature.
If the other reviewers recommend the paper for publication, the reviewer will follow the opinion.